



# Rainfall drives atmospheric ice nucleating particles in the maritime climate of Southern Norway

Franz Conen[1], Sabine Eckhardt[2], Hans Gundersen[2], Andreas Stohl[2], Karl Espen Yttri[2]

[1]Department of Environmental Sciences, University of Basel, Basel, Switzerland
[2]NILU – Norwegian Institute for Air Research, Kjeller, Norway

*Correspondence to*: Franz Conen (franz.conen@unibas.ch)

**Abstract.** Ice nucleating particles active at warm temperatures (e.g. -8 °C; $INP_{-8}$) can transform clouds from liquid to mixed-phase, even at very small number concentrations (<10 m$^{-3}$). Over the course of 15 months, we found very similar patterns in weekly concentrations of $INP_{-8}$ in $PM_{10}$ (median = 1.7 m$^{-3}$, maximum = 10.1 m$^{-3}$) and weekly amounts of rainfall (median =
28 mm, maximum = 153 mm) at Birkenes, southern Norway. Probably, $INP_{-8}$ were aerosolised locally by the impact of raindrops on plant, litter and soil surfaces. Major snowfall or heavy rain onto snow-covered ground were not mirrored by enhanced numbers of $INP_{-8}$. Further, source receptor sensitivity fields obtained from transport model calculations for large and small numbers of $INP_{-8}$ differed more in their likelihood to provide precipitation to southern Norway, than in the proportion of land cover or landuse type. In $PM_{2.5}$ we found consistently about half as many $INP_{-8}$ as in $PM_{10}$. From mid-
May to mid- September, $INP_{-8}$ correlated positively with the fungal markers arabitol and mannitol, suggesting that $INP_{-8}$ may consist largely of fungal spores. In the future, warmer winters with more rain instead of snow may enhance airborne concentrations of $INP_{-8}$ during the cold season in southern Norway and in other regions with a similar climate.

## 1 Introduction

Most precipitation in the mid-latitude and polar regions is linked to ice formation in clouds (Field and Heymsfield, 2015).
Biological particles catalyse the freezing of supercooled cloud droplets at temperatures between -1 and -15 °C, whereas other particles (e.g. mineral dust, soot) are active at colder temperatures (Després et al., 2012; Murray et al., 2012). The ice particles thus formed grow to snow flakes through vapour deposition and, more rapidly, through riming and aggregation. A few initial ice particles (< 10 m$^{-3}$) catalysed at -8 °C near the cloud top can rapidly glaciate a shallow supercooled cumulus (Mason 1996; Crawford et al., 2012). This is possible through explosive ice multiplication by riming and splintering graupel
pellets between -3 and -8 °C (Hallett and Mossop, 1974). Therefore, ice nucleating particles active at -8 °C or warmer ($INP_{-8}$) could strongly influence precipitation development, despite their usually small number concentration in the atmosphere compared to mineral dust or soot particles. The few atmospheric data on $INP_{-8}$ suggest that vegetated lands are stronger sources than deserts (Conen et al., 2015), and that rainfall triggers the aerosolisation of such INP from forest (Hara et al., 2016). Aerosolisation is probably due to the mechanical impact of raindrops on surfaces hosting organisms that, as a





whole or in parts, can serve as INP. Increased concentration during and after rainfall has also been demonstrated for INP active at colder temperatures (-15 °C) by Bigg and Miles (1964) at 24 sites in Australia, and by Huffman et al. (2013), Tobo et al. (2013) and Prenni et al. (2013) at a site in Colorado, USA. Investigations at the latter site were accompanied by detailed characterisation of aerosolised particles regarding their size distribution, fluorescence, morphology and biological

origin. Huffman et al. (2013) suggested that *"[f]ollow-up studies in other environments shall elucidate whether the observed rain-related bioaerosol increase is a common feature of terrestrial ecosystems or specific for the investigated semi-arid environment"*. Thus we took this suggestion as a starting point to investigate rainfall effects in the maritime climate of Southern Norway.

## 2 Material and Methods

Unlike previous studies, we continuously sampled aerosol particles over the course of 15 months, from October 2013 to December 2014, on a filter replaced once a week. The low time resolution of one week provided for a large sampled volume, which enabled the reliable detection of $INP_{-8}$ in all samples.

### 2.1 Site description

The Birkenes Observatory (58°23'N, 8°15'E, 219 m asl) is situated approximately 20 km from the Skagerrak coast in

southern Norway (Figure 1), and is located on a minor hilltop in an undulating terrain, which allows for efficient ventilation and air mass mixing. Kristiansand (61 000 inhabitants, 2016) is the nearest city, located 25 km south/south-west of the station. The Observatory is located in the Boreo-nemorale zone with mixed coniferous and deciduous trees accounting for 65% of the land use near the site. Meadows and low intensity agricultural areas account for 10% each, whereas 15% are freshwater lakes. Birch is the most common deciduous tree in the area around the Observatory, for which budding starts in

late April and shedding of leaves is over by mid-October.

By its proximity to the coast and low altitude, the Birkenes Observatory experiences a maritime climate, with relatively mild winters ($T_{mean\ Jan\ -\ Feb\ 2014}$ = -0.3 °C) and moderately warm summers ($T_{mean\ Jun\ -\ Aug}$ 2014 = 15.6 °C). The annual amount of precipitation around Birkenes was 2077 mm in 2014, about 1.5 times the normal (1961-1990) amount. Of this, 12%

precipitated as snow. Temperature exceeded in 2014 the norm by 2-3 °C (MET, 2015). The prevailing wind direction is from the west and the south west, occasionally with quite high wind speeds. Hence, the Observatory is situated downwind of major emission regions in continental Europe.

### 2.2 Aerosol sampling

Ambient aerosol filter samples were collected as part of the Norwegian national monitoring programme (Aas et al., 2015),

using two Kleinfiltergerät low-volume samplers with a $PM_{10}$ and a $PM_{2.5}$ inlet to collect aerosols on prefired (850 °C; 3 h)





single quartz fiber filters (Whatman QM-A; 47 mm in diameter). Both samplers operated at a flow rate of 38 L min$^{-1}$, corresponding to a filter face velocity of 47.3 cm s$^{-1}$. Filters were conditioned at 20 ± 1°C and at 50 ± 5% RH (relative humidity) for 48 h before and after exposure. Filters were kept in petri slides for transportation and storage. Post conditioning, filters were stored at 4 °C for approximately one month for subsequent analysis of OC/EC, then at -18 °C prior

to analysis of arabitol and mannitol.

### 2.3 Analysis of ice nucleating particles

Number concentrations of INP$_{-8}$ on PM$_{10}$ and PM$_{2.5}$ filter samples were determined with 108 punches (1.0 mm diameter) from each filter. Each punch was immersed in Milli-Q water (0.1 ml) in a tube (1.5 ml, Eppendorf Safe-Lock), cooled from -4 to -12 °C (0.3 °C min$^{-1}$) in a cold bath (Lauda, model RC6). The number of frozen tubes were counted every 1 °C

temperature step to calculate the number concentration of INP in sampled air (Conen et al., 2012). The punches from 10 filters in each size fraction were tested a second time after they had been immersed for 10 min in a water bath at 90 °C. We also tested 24 field blanks the same way. Only one had a small positive signal (0.08 INP$_{-8}$ m$^{-3}$) and 23 were negative at -8 °C, thus we did not do a blank correction. As we have analysed punches of each filter in two lots of 54, we can estimate the uncertainty of our procedure. The mean deviation of INP$_{-8}$ derived from single lots of 54 punches, from that of both lots

taken together (108 punches), was 22%, 19%, 13%, and 12% for INP$_{-8}$ < 1, 1 to 2, 2 to 4, and > 4 m$^{-3}$, respectively.

### 2.4 Analysis of arabitol and mannitol

Concentrations of arabitol and mannitol in PM$_{10}$ filter samples were determined using Waters Acquity ultra-performance liquid chromatography (UPLC) in combination with Waters Premier XE high-resolution time-of-flight mass spectrometry (HR-TOFMS) operated in the negative electrospray ionization (ESI-) mode: resolution > 10000 FWHM (Full width half

maximum). The analytical methodology is based on that described by Dye and Yttri (2005) for monosaccharide anhydrides, deviating from the original one only by choice of the column (2.1 x 150 mm HSS T3, 1.8 µm, Waters Inc.). Arabitol and mannitol were identified on the basis of retention time and mass spectra of authentic standards (ICN Biomedicals). Response factors for arabitol and mannitol were calculated from external standards. Isotope-labelled standards of levoglucosan ($^{13}$C-levoglucosan, 98%, Cambridge Isotopic Laboratories) were used as internal recovery standard.

### 2.5 FLEXPART

FLEXPART is a Lagrangian particle dispersion model (Stohl et al., 1998, 2005) and is used to investigate the origin of air masses and their potential for emission uptake. The model is driven by 1x1 degree operational meteorological data from the European Centre for Medium Range Weather Forecast (ECMWF) with 3 hourly temporal resolution and 137 vertical levels. The model calculates the trajectories of tracer particles using the interpolated mean winds plus random motions representing

turbulence and moist convection. The particles are subject to dry and wet deposition, the latter of which is described in detail in Grythe et al. (2016).



For this study, FLEXPART was run 20 days backward in time with a black carbon tracer, which experiences wet and dry deposition. Black carbon was used as a proxy for INP, as both are susceptible to dry and wet scavenging. For the exact time interval each filter sample was collected, 400.000 particles were released. The FLEXPART output is a potential emission sensitivity in units of seconds. It quantifies the impact of potential emissions on the aerosol concentration at the

measurement site. If multiplied with known emission fluxes, the aerosol concentration at the receptor is obtained. Since emissions of INP are not known, we use arbitrary constant emission densities for the different landuse types, based on the International Geosphere Biosphere Program (IGBP) data (Belward, 1999) at 1x1 degree resolution. The landuse categories used are: urban, agriculture, range land, deciduous forest, mixed forest, coniferous forest, water, desert, wetland, agriculture/range land, rocky areas, snow and rainforest. Our procedure thus quantifies the relative impact that INP

emissions in these landuse types would have on the INP concentration in a measurement sample.

### 3 Results and Discussion

#### 3.1 Time series of $INP_{-8}$ and precipitation

Our observations revealed a general seasonal pattern. Values of $INP_{-8}$ in $PM_{10}$ were mostly $< 2$ m$^{-3}$ during spring and summer, and $> 4$ m$^{-3}$ over the course of several weeks during autumn. Values decreased with snowfall in winter and were on

average 1.2 m$^{-3}$ as long as the ground was covered by snow (Fig. 2). Elevated $INP_{-8}$ levels of short duration in spring and summer were associated with rainfall exceeding 20 mm per week. Similar rates of snowfall (January 2013) were not accompanied by additional $INP_{-8}$.

Equally, Hara et al. (2016) saw enhanced concentrations of $INP_{-7}$ during rainfall, but not during snowfall at Kanazawa, a

coastal city in Japan. Huffman et al. (2013) suggested that rainfall may trigger the release of INP from vegetation through its mechanical impact. Such impact happens on leaves, but also on other surfaces, such as twigs, stems, soil, and on leaf litter covering the forest floor. If leaves on trees were the only relevant source of INP, we should have seen a marked decrease in atmospheric concentrations once leaves were shed by mid-October. No change of this kind was discernible in the autumn of 2013 or 2014. Numbers of INP on shed leaves increase substantially upon decay (Vali et al., 1976). At first sight, it seems

unlikely that INP produced on the forest floor are transported above the canopy. However, once trees are defoliated, raindrops hit the ground with less interception by canopies, and turbulent momentum penetrates more easily into the sub-canopy layer (Staebler and Fitzjarrald, 2005). Continued occurrence of large numbers of airborne $INP_{-8}$ during the defoliated period suggests that the reduced likelihood of aerosolisation from the forest floor is largely compensated by the increased source strength of INP due to the decay of the shed leaves. Throughout the year, the increase in $INP_{-8}$ with weekly rainfall

occurs despite the fact that scavenging of such particles is also enhanced by rain.





An increase or decrease in the amount of rainfall from one week to the other was followed in 76% of all cases by a change in $INP_{-8}$ in the same direction (2013 and 2014, excluding weeks with snowfall; n = 58). However, the change in $INP_{-8}$ was not always proportional to the change in rainfall. In particular, large amounts of rainfall during February 2014 had a relatively small effect. Snowfall in January had left the ground covered with one metre of snow, which diminished during February

and had disappeared completely in the second half of March. Pearson's $r$ for the correlation between amount of rainfall and $INP_{-8}$ has a value of 0.45 for all weeks without snowfall (n = 61) and 0.53 for all weeks with neither snowfall nor snow on the ground (n = 55). Both correlations are statistically significant (p < 0.01). There are at least two reasons for deviations: First, depending on rain intensity and raindrop size, the same amount of weekly rainfall can have a very different impact on aerosolising microorganisms from vegetation (Paul et al., 2004). Second, presence and density of IN-active material on

surfaces hit by raindrops may change with season or on shorter time scales. Considering both issues, the time course of $INP_{-8}$ mirrors that of rainfall surprisingly well (Fig. 2).

**3.2 Potential source regions**

We summarised source receptor sensitivity (SRS) fields for situations with > 4 $INP_{-8}$ $m^{-3}$ and when $INP_{-8}$ were < 4 $m^{-3}$ in $PM_{10}$ (Fig. 3). When $INP_{-8}$ were > 4 $m^{-3}$, there was strikingly less influence from the northeast (Fennoscandia, Norwegian

Sea, Barents Sea, Northwest Russia, Siberia), but a larger influence from the lower latitudes (Southwestern Europe). Air masses from the northeast arrive at Birkenes mainly with high pressure systems and bring no or little rain. Large amounts of rain arrive with cyclones of the North Atlantic stormtrack. Although high $INP_{-8}$ concentrations are also associated with enhanced transport from Northern Africa, this transport signal is due only to a few cases and unlikely to cause the higher $INP_{-8}$ levels. Furthermore, the Saharan Air Layer sampled on Tenerife over the course of a year, and analysed by the same

method as we used here, was previously found to contain no more than 1 $INP_{-8}$ $m^{-3}$, even during dust storms (Conen et al., 2015). Overall, there was only a minor difference in the proportion of influence from land surfaces between situations with higher or lower values than 4 $INP_{-8}$ $m^{-3}$ (48% and 51%, respectively), and even smaller differences in land cover type. The small difference in SRS fields supports the idea that it is rain and its impact on aerosolising $INP_{-8}$ locally that drives temporal variation in $INP_{-8}$ concentrations at Birkenes.

**3.3 Size of $INP_{-8}$**

At seven other sites across the Northern Hemisphere, Mason et al. (2016) found 38% to 90% of $INP_{-15}$ collected with a cascade impactor to be larger than 2.5 µm (mean = 62%, s.d. = 20%). In the present study, $INP_{-8}$ were equally distributed amongst the fine ($PM_{2.5}$) and the coarse fraction ($PM_{10-2.5}$) of $PM_{10}$ (Fig. 4). Such a 50-50% distribution can be expected, if there is a unimodal, normal distributed peak of $INP_{-8}$, which peaks around 2-3 µm aerodynamic diameter. Some findings

support this presumption. Schumacher et al. (2013) measured continuously during 18 months the size distribution of fluorescent biological aerosol particles (FBAP) at a site in southern Finland, 960 km to the East of Birkenes. Their results





showed a dominant, often extremely narrow mode between 2.5 and 3.0 µm. Similar observations at a forest site in Colorado revealed rain-induced increases in FBAP with a peak at 2.0 µm (Schumacher et al., 2013).

Huffman et al. (2013) have demonstrated for the site in Colorado a close link between FBAP and INP. They saw rainfall to immediately increase the number of airborne biological particles with a size of 2-3 µm, together with an increase in $INP_{-15}$. In addition, larger particles (4-6 um), comprising INP, appeared several hours later and lasted for up to 12 hours after rainfall had ceased. Huffman et al. (2013) were able to identify several IN-active species from aerosol samples collected during rainfall, including two fungi previously unknown to be IN-active that produced spores between 1 and 4 µm in geometric size. Strong correlations between FBAP and fungal markers, such as arabitol, strongly supported the notion that most FBAP

aerosolised during rain were of fungal origin (Gosselin et al., 2016).

Another important source of FBAP in $PM_{10}$ are pollen. Pollen from birch, the most abundant deciduous tree around the Birkenes Observatory, has an aerodynamic diameter of 20 µm (Efstathiou et al., 2011). Smaller fragments of pollen are generated by osmotic rupture when pollen grains get wet. These fragments are as IN-active as intact grains (Pummer et al.,

2012). On rainy days their abundance increases in the fine fraction (Rathnayake et al., 2017). However, we can exclude a major contribution of pollen-derived INP, because exposure to 90 °C deactivated on average > 93% of $INP_{-8}$ in our assays. The INP of bacteria and most fungi are proteins and denatured at this temperature. INP from pollen or fractured pollen are most likely polysaccharides and would have remained active after heating to 90 °C (Pummer et al., 2012, 2015).

Recently, Wang et al. (2016) described a mechanism that generates airborne soil organic particles (ASOP) of sub-micron size by air bubbles bursting at the water-air interface of impacted raindrops. Soil organic matter can harbour large numbers of $INP_{-8}$ (Schnell and Vali, 1972; Conen et al., 2011; O'Sullivan et al., 2015; Hill et al., 2016). Therefore, some of the $INP_{-8}$ in the fine fraction at Birkenes might be of that kind.

### 3.4 Time series of arabitol and mannitol

Arabitol and mannitol had very similar temporal pattern (Fig. 5). Their concentrations were low from January to mid-April, then increased, and remained enhanced throughout summer. During the warmer part of the year, from mid-May to mid-September, they correlated significantly with $INP_{-8}$ (mannitol: r = 0.72, p < 0.01; arabitol: r = 0.48, p < 0.05). These correlations support the above-mentioned notion that FBAP and $INP_{-8}$ may be mainly of fungal origin, at least during the warmer part of the year. Another hint in this direction comes from measurements about 300 km north east of Birkenes.

There, in summer, arabitol and mannitol had a unimodal size distribution peaking between 2 and 4 µm aerodynamic diameter (Yttri et al., 2007), which coincides with our interpretation of the 50-50% distribution of $INP_{-8}$ amongst $PM_{2.5}$ and $PM_{10-2.5}$. Concentrations of the fungal markers increased towards the end of three rather dry weeks in September and reached their annual maxima with intensive rainfall at the beginning of October, followed one week later by a sevenfold increase in $INP_{-8}$.





Changes in INP$_{-8}$ continued to follow those of the fungal markers with a delay of one week until the beginning of December. During December, INP$_{-8}$ remained elevated while concentrations of arabitol and mannitol decreased markedly. All reached their minimum with snowfall in the last week of the year.

Assuming 1.2 pg arabitol and 1.7 pg mannitol per fungal spore (Bauer et al., 2008), we estimate for the period from mid-May to mid-September average spore concentrations of 4.6 and 5.8 x 10$^3$ m$^{-3}$. The average concentration of INP$_{-8}$ during the same period was 1.6 m$^{-3}$. If all INP$_{-8}$ were spores, there would have been 2.8 or 3.5 x 10$^{-4}$ INP$_{-8}$ per spore, which is in the upper range of values reported by Morris et al. (2013, Fig. 1) for urediospores of rusts. For the last three months of the year, the average INP$_{-8}$ to spore ratio was about twice as large. Since INP$_{-8}$ were no longer directly related to fungal spore markers
during this period, it might be that bacteria contributed more to the total number of INP$_{-8}$ than spores did during the warmer months before. The expression of ice nucleation activity in bacteria is favoured by cold temperatures and nutrient limitation (Nemecek-Marshall et al., 1993). Hence, when similar numbers of bacteria are aerosolised by the same amount of rainfall, they likely contribute larger numbers of INP$_{-8}$ during the cold season, than during the warmer months.

**4 Conclusion**

Abundant rainfall in the maritime climate of Southern Norway drives the near surface concentration of INP$_{-8}$ across all seasons. Concentrations of INP$_{-8}$ increase with amounts of rain. Most airborne INP$_{-8}$ are probably aerosolised locally through the impact of raindrops onto surfaces hosting INP-producing microorganisms. Snowfall has no such effect. When trees are defoliated between October and April, decaying leaf litter on the ground constitutes a likely INP source. During this time,
snow cover on the ground strongly reduces such INP aerosolisation by rainfall.

Rain-released INP$_{-8}$ are equally distributed amongst the fine (PM$_{2.5}$) and the coarse fraction (PM$_{10-2.5}$) of PM$_{10}$ Sensitivity to heat treatment (90 °C) suggests bacterial or fungal sources, not pollen. The assumption of fungal sources is further supported by some similarities in the temporal pattern of INP$_{-8}$ and the fungal markers arabitol and mannitol, but only during the
warmer part of the year. During the colder part of the year, bacterial contributions may be more important than fungal sources.

In general terms, we expect similar relations between rainfall and warm temperature INP in other coastal regions with a comparable climate and ecosystem, such as the Pacific coasts of Canada and Chile, Japan, and New Zealand. Global
warming may lead to shorter periods of snow cover on the ground and a greater proportion of precipitation falling as rain instead of snow. These changes would probably result in larger airborne concentrations of INP$_{-8}$ during the cold season. Whether they have an effect on cloud development in these regions remains an interesting question for further studies.





## 5 Acknowledgements

The PM$_{10}$ and PM$_{2.5}$ filter samples used for measurements of ice nucleating particles in the present study were analysed from filters obtained as part of the Norwegian national monitoring program (Aas et al., 2015). Precipitation and snow data were obtained from the Norwegian Meteorological Institute through eKlima.

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

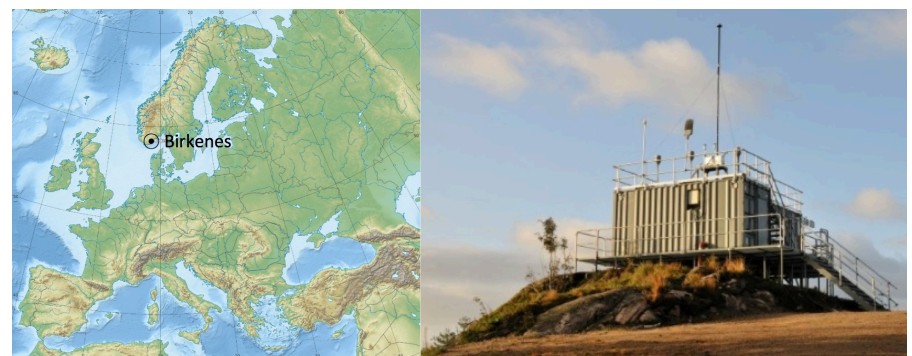

**Figure 1: Location of Birkenes Observatory in Europe (left) and a view of the Observatory (right).**

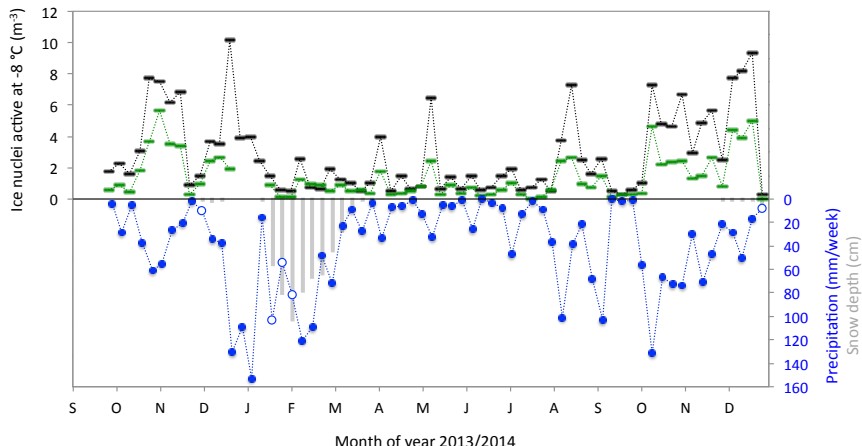

**Figure 2: Time course of INP$_{-8}$ in the PM$_{10}$ (black) and PM$_{2.5}$ (green) particle size fractions. Peaks in INP$_{-8}$ largely mirror peaks in precipitation (blue; inverse scale). Snow (open circles) or small amounts of rain are accompanied by very small numbers of INP$_{-8}$ in the same week. Significant amounts of snow on the ground (grey bars) seem to attenuate the increase in INP$_{-8}$ with intense rainfall. Total amount of precipitation in 2014 was 2077 mm (entire period shown: 2640 mm). Precipitation and snow cover data are the averages of three stations operated by the Norwegian Meteorological Institute in the municipality of Birkenes.**





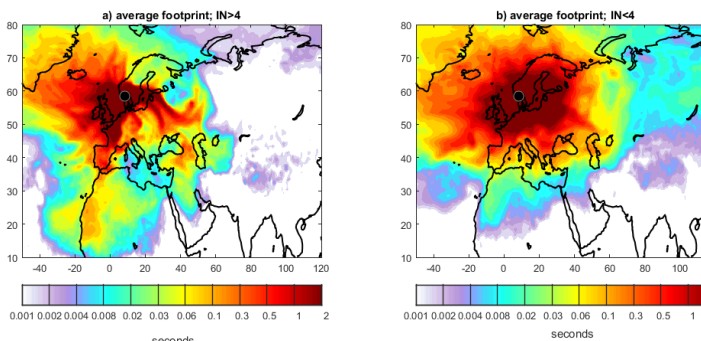

**Figure 3: Source receptor sensitivity (SRS) fields for situations with > 4 INP$_{-8}$ m$^{-3}$ (left) and when INP$_{-8}$ were < 4 m$^{-3}$ (right).**

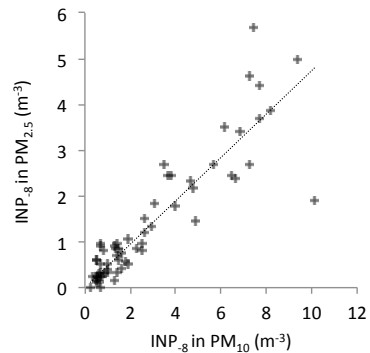

**Figure 4: Ice nucleating particles active at -8 °C in PM$_{2.5}$, relative to their number in PM$_{10}$. The slope of the regression line has a value of 0.47.**





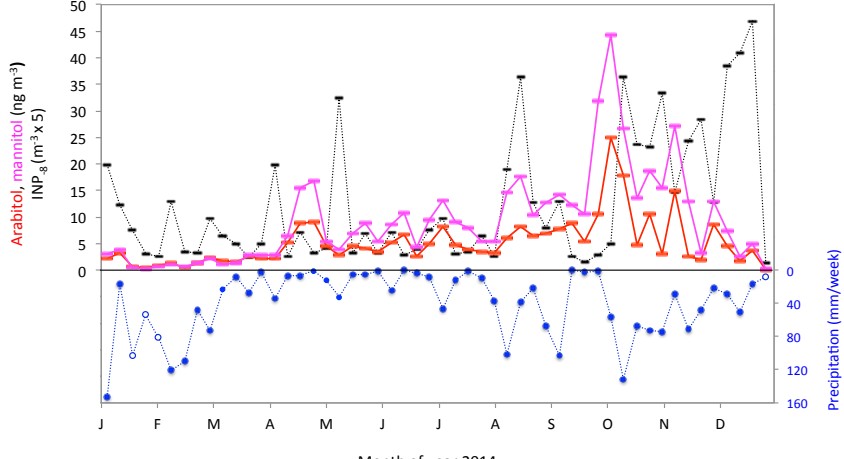

**Figure 5: Weekly concentration of arabitol (red) and mannitol (magenta). The time courses of INP$_{-8}$ in PM$_{10}$ (black; x 5, to fit onto the same scale) and precipitation (blue) are copied from Fig. 1 to facilitate direct comparison.**