# Peer review of "Rainfall drives atmospheric ice nucleating particles in the maritime climate of Southern Norway"

_Atmospheric Chemistry and Physics, 2017_

## Referee Comment (RC1) · Anonymous Referee #1 · 2 May 2017

The authors show the results of monitoring of ice nucleating particles (INPs) active at a relatively warm temperature (-8°C) in PM10 and PM2.5 at a coastal site in southern Norway. They also discuss possible linkages of INPs and local meteorological events. Despite low time resolution and limitation of available temperature (weekly INP data at -8°C), this work provides rare and very valuable datasets, since there are few studies that have reported seasonal variations of atmospheric INPs based on continuous long-term monitoring. However, I thought that more detailed discussion is necessary to suggest that INPs active at -8°C may consist largely of fungal spores. I would therefore recommend this manuscript be published after several revisions.

Specific comments:

1) The wording "maritime climate (e.g., title)" may not be appropriate. This work is based on the measurements at a coastal site, but I didn't think that the measurements have been conducted in remote marine boundary layer. For example, the authors described that "probably, INP-8 were aerosolised locally by the impact of raindrops on plant, litter and soil surfaces (Page 1, Lines 10-11)". This obviously indicates the influence of terrestrial origin and not marine origin. To avoid misleading, I would strongly suggest avoiding the use of the wording "maritime climate" in the main text and considering a more appropriate title.

2) It is a little difficult to understand the motivation of measurements of arabitol and mannitol. Although there is the description of "fungal markers arabitol and mannitol (Page 1, Line 15)" in the abstract, the authors should clearly explain why arabitol and mannitol can be regarded as fungal biomarkers in Section 2.4. In addition, in the first paragraph of Section 3.4, the authors may need to briefly describe the reason why they decided to use arabitol and mannitol data.

3) The results of measurements of INPs after heating to 90°C should be presented more clearly. The authors conclude that "sensitivity to heat treatment (90°C) suggests bacterial or fungal sources, not pollen (Page 7, Lines 22-23)". For this results, however, there is only a very short description that "exposure to 90 °C deactivated on average > 93% of INP-8 in our assays (Page 6, Line 16)". If "the punches from 10 filters in each size fraction were tested a second time after they had been immersed for 10 min in a water bath at 90 °C (Page 3, Line 10-11)", it is probably important to show the results in a figure and/or table (and seasonal variations of INP-8 after heat treatment if possible).

4) It is hard to understand why the authors consider that "INP-8 may consist largely of fungal spores during the warm part of the year (Page 1, Line 15)" and "bacterial contributions may be more important than fungal sources during the colder part of the year (Page 7, Line 25, 26)". I assume that this is only based on the results that "from mid-May to mid-September, INP-8 correlated positively with the fungal markers arabitol and mannitol (Page 1, Line 15; Page 6, Lines 26-27)". This result may suggest that fungal
spores are a potential important source of INPs from mid-May to mid-September, but I think that it is still impossible to rule out the possibility that the contribution of bacteria and soil organic particles were also significant. Did you try any other approaches to support the authors' hypothesis? For example, did you evaluate the relationship of INP-8 with other markers (e.g., bacteria, soil organic particles) from mid-May to mid-September?

5) It is a little hard to understand that "from mid-May to mid-September, INP-8 correlated positively with the fungal markers arabitol and mannitol (Page 1, Line 15)" only based on Figure 5. For example, could you show the results (additional figures like Figure 4 or tables) comparing INP-8 with arabitol and mannitol measured in different seasons (e.g., the period of mid-May to mid-September vs. other periods; spring vs. summer vs. autumn vs. winter)?

Technical corrections:

6) Page 2, Line 5: [f]ollow-up => follow-up?

7) Please explain the definition of INP-8 more clearly. Is it ice nucleating particles "active at -8°C or warmer (Page 1, Line 25)" or "active at -8°C"?

8) It is a little difficult to see the data on precipitation and/or snow depth in Figures 2 and 5. I would like to suggest that the values would increase from bottom to top if there are no special reasons.

9) What is the value of 0.47 in Figure 4 (r or r2)? In addition, I would like to suggest that the authors would indicate the equation of the regression line, since they noted that "in PM2.5 we found consistently about half as many INP-8 as in PM10 (Page 1, Line 14)".

10) Is Figure 3 FLEXPART output? If so, please describe it in Section 3.2 and/or the caption of Figure 3. In addition, I would like to suggest including the explanation of why the unit of potential emission sensitivity is seconds (Page 4, Lines 3-4).

---

## Referee Comment (RC2) · JA Huffman (Referee) · 8 Jul 2017

Conen et al. submitted a manuscript for review titled "Rainfall drives atmospheric ice nucleating particles in the maritime climate of Southern Norway." The manuscript compares 15 months of measurements of ice nucleating particles (at -8C), two molecular tracers (arabitol and mannitol), and rainfall data to present observations about INP behavior. The authors suggest that INP were likely to have local sources and are linked to rainfall, because of the evidence that INP concentrations correlated with rain. Further, they state that correlations with molecular tracers suggest INP "may consist largely of fungal spores." The manuscript presents interesting environmental data linking warm

temperature INP with rainfall and two commonly used molecular tracers. The region sampled is also not over-represented in literature and so provides some atmospheric perspective on this region, including possible parallels with other similar regions of the world, as mentioned. In general, I support the publication of this manuscript, but there is some work that I suggest be done before it is accepted. The analysis and evidence that the observed INP are fungal in nature are both relatively thin and should be improved. I list some suggestions for specific additions below, including some possibilities for added discussion and some suggestions to add to quantitative evidence. These statements are meant to suggest possible areas of improvement, but are not necessarily meant to be comprehensive.

General comments: Abstract: "INP(-8) correlated positive with the fungal markers arabitol and mannitol, suggesting that INP may consist largely of fungal spores." I think the confidence implied by this conclusions is somewhat over-stated. The evidence shown suggests to me that the INP have a source that is at least correlated with arabitol and mannitol, but this does not necessarily mean that the INP are spores themselves. The observations could also be explained if INP and fungal spores are co-emitted by a similar process or are somehow physically related to one another. A lot of evidence has suggested many fungal spores are not IN-active (e.g. Iannone et al., 2011) while other (i.e. rust spores from the cited Morris et al. paper) are IN-active at high temperatures. There is enough complexity in this conversation, that I think some discussion of these differences should be mentioned and the overall confidence in the implications that fungal spores are the source of INP should be scaled down a bit.

The discussion about molecular tracers should also be extended somewhat. For example, arabitol and mannitol are commonly used as tracers for fungal spores, but not without complications. One important thing to mention here is that these specific tracers are typically used as tracers for wet-discharge spores, but only poorly relate to other types of spores (i.e. dry discharge spores like Cladosporium that can be ubiquitous and a large fraction of spore mass). How might this understanding impact the

conclusions that are being drawn here? I am aware that the general knowledge linking these tracers with ice activity is low, and so it is unreasonable to require any kind of a quantitative link between known ice fungal ice nucleators and the amount of arabitol or mannitol they release, but I suggest at least mentioning some of the uncertainties that come along with the analysis and assumptions as presented.

In general, I think Section 3.2 and Figure 3 need more detailed discussion and explanation to help a reader not experienced with this type of analysis. Can you explain what the z-scale implies from this figure and how it relates to the brief observations you make? It's hard to know how much to make of the summary observations reported. How much of this is a function of different averaging times that may lead to random differences? If this is an important piece of evidence, is there some statistical treatment that can be applied here? Flipping back and forth between the comments and the figure I can follow the reasoning of the trends mentioned, but it is hard to know whether the "striking" comment (P5 L14) is stronger than I would have stated – at least having briefly looked at the differences. If the authors are confident of the strong difference, I suggest improving the evidence for that distinction. In contrast to this, however, the last sentence in Section 3.2 essentially says that the authors think the effects are local, which implies to me that Figure 3 should provide evidence *against* long-distance sources, right? This goes back to how Figure 3 should be interpreted as striking differences between high and low INP concentration.

What happens if you do correlations of the traces in Figures 2 and 5? A lot of the observations come down to qualitative comparisons of these traces, but it is hard to know what this means quantitatively. I think this is one obvious area that could easily improve the manuscript. Without evidence beyond the visual trends presented, it is hard to know how much to make of the possible co-variance. As a simple addition, I would also add the R2 value (or something similar) to Figure 4.

Looking at Figure 5, it seems that there is a one-point lag in INP behind arabitol and mannitol during approximately October and November. Do you think this is real? If so,

what might be causing this? Or is it just a figure illusion and statistical aberration.

P7 L10: The statement here is that "Since INP were no longer directly related to fungal spore markers during this period, it might be that bacteria contributed more to the total number . . .". Another possibility is that the type of spores being released are of a different variety and are just less efficient at producing IN-active.

Bigg, Soubeyrand, and Morris recently published a paper reporting long-term statistical correlations between ice nuclei and rainfallin Australia (Bigg et al., 2015). I think reference to this work would be appropriate here, probably in the conclusions.

P6 L1: The authors cite Schumacher et al. as having observed a 2.5 – 3.0 um mode of fluorescent particle during 18-months of study in Finland. That paper also mentions a prominent decrease in fluorescent particles during snow-covered periods, which qualitatively matches some of the observations shown here.

Is snowfall poor at launching INP because of snow-covered vegetation and soil or also because the kinetic velocity at which the drops fall does not kick up material? Some recent papers on rainfall velocity and particle ejection could be cited and discussed here (e.g. P7 L18).

P7 L23: Are the heat treatment properties of fungal proteins the same as bacterial proteins? I think of spores as relatively robust, and so I wonder if it is possible for some fraction of spore material to remain active, whereas the fraction for bacteria goes to zero? In any case, I think the evidence for these arguments should be stronger.

P 6 L12: Another paper by Maninnen et al. (2014) shows seasonal trends in pollen and fungal spores at the boreal Hyttiala site in Finland and they also break the analysis down into PM mass <2.5 um and >10 um. While not at the same land-use type, these measurements may (or may not) be useful for broad comparison here.

Minor technical comments: P1 L10: Move placement of "probably" to "INP were probably aerosolized . . ." P1 L12-14: I though this sentence was confusing and could use

a revision to make the point clearer. P1 L22: snowflake is one word P4 L7 , L8, L10: "Landuse" should be two words P5 L19: Specifically mention that Tenerife is off the W coast of northern Africa

References: Bigg, E. K., Soubeyrand, S., and Morris, C. E.: Persistent after-effects of heavy rain on concentrations of ice nuclei and rainfall suggest a biological cause, Atmos. Chem. Phys., 15, 2313-2326, 2015. Iannone, R., Chernoff, D. I., Pringle, A., Martin, S. T., and Bertram, A. K.: The ice nucleation ability of one of the most abundant types of fungal spores found in the atmosphere, Atmospheric Chemistry and Physics, 11, 1191-1201, 2011. Manninen, H. E., Bäck, J., Sihto-Nissilä, S.-L., Hufffman, J. A., Pessi, A.-M., Hiltunen, V., Aalto, P., Hidalgo, P. J., Hari, P., Saarto, A., Kulmala, M., and Petäjä, T.: Patterns in airborne pollen and other primary biological aerosol particles (PBAP), and their contribution to aerosol mass and number in a boreal forest, Boreal Environmental Research, 19, 383-405, 2014.

---

## Author Response (AR1)

MS No.: acp-2017-285

Response to Anonymous Referee #1
* * *
To facilitate reading we use different fonts for (1)comments from Referee, (2) our response, *(3) the changes we made to the manuscript.*
* * *
The authors show the results of monitoring of ice nucleating particles (INPs) active at a relatively warm temperature (-8° C) in PM10 and PM2.5 at a coastal site in southern Norway. They also discuss possible linkages of INPs and local meteorological events. Despite low time resolution and limitation of available temperature (weekly INP data at -8° C), this work provides rare and very valuable datasets, since there are few studies that have reported seasonal variations of atmospheric INPs based on continuous longterm monitoring. However, I thought that more detailed discussion is necessary to suggest that INPs active at -8 °C may consist largely of fungal spores. I would therefore recommend this manuscript be published after several revisions.

We thank Referee #1 for having taken the time to read our manuscript and for providing valuable feedback on how to improve it.

The claim that fungal spores make up a majority of INP$_{-8}$ at Birkenes seems too weak to stand the discussion. It was also questioned by Alex Huffman, the second Referee. Additional reading related to this issue led us to a more differentiated view and according changes to the manuscript.

Specific comments:

1) The wording "maritime climate (e.g., title)" may not be appropriate. This work is based on the measurements at a coastal site, but I didn't think that the measurements have been conducted in remote marine boundary layer. For example, the authors described that "probably, INP-8 were aerosolised locally by the impact of raindrops on plant, litter and soil surfaces (Page 1, Lines 10-11)". This obviously indicates the influence of terrestrial origin and not marine origin. To avoid misleading, I would strongly suggest avoiding the use of the wording "maritime climate" in the main text and considering a more appropriate title.

With "maritime climate" we meant to indicate that the climate at the observatory is shaped by the proximity of the sea. "Coastal climate" is equally valid and avoids misunderstanding.

*We replaced "maritime climate" with "coastal climate" throughout the manuscript (incl. title).*

2) It is a little difficult to understand the motivation of measurements of arabitol and mannitol. Although there is the description of "fungal markers arabitol and mannitol (Page 1, Line 15)" in the abstract, the authors should clearly explain why arabitol and mannitol can be regarded as fungal biomarkers in Section 2.4. In addition, in the first paragraph of Section 3.4, the authors may need

```
to briefly describe the reason why they decided to use arabitol and
mannitol data.
```

Sorry for the operational blindness. We fully agree that more explanation is necessary.

*We added a short sentence to the beginning of section 2.4: "Arabitol and mannitol have previously been identified as amenable tracers of fungal spores (Bauer et al., 2008)." (page 3, line 20).*
*Greater detail seemed more appropriate under 'Results and Discussion', at the beginning of Section 3.4. There, we added: "Arabitol and mannitol serve as carbohydrate stores in fungal spores. Their ambient air concentration has been found to correlate well with number concentrations of airborne fungal spores (Bauer et al., 2008), but not necessarily with ergosterol (Burshtein et al., 2011), a dominant sterol in most fungi (Weete et al., 2010). It seems that arabitol and mannitol are specifically associated with spores released under moist conditions, as occur in forests during nighttime (Zhu et al., 2016)." (page 7, lines 6-9).*

```
3) The results of measurements of INPs after heating to 90 °C should
be presented more clearly. The authors conclude that "sensitivity to
heat treatment (90 °C) suggests bacterial or fungal sources, not
pollen (Page 7, Lines 22-23)". For this results, however, there is
only a very short description that "exposure to 90 ° C deactivated on
average >93% of INP-8 in our assays (Page 6, Line 16)". If "the
punches from 10 filters in each size fraction were tested a second
time after they had been immersed for 10 min in a water bath at 90 °C
(Page 3, Line 10-11)", it is probably important to show the results
in a figure and/or table (and seasonal variations of INP-8 after heat
treatment if possible).
```

For nine of the 10 samples tested a second time after heat treatment we can only provide an upper estimate for heat resistant INP because their number was below detection limit. Therefore, there is no seasonal structure we could show.

*We now explain this issue in more detail in the 3rd paragraph of section 3.3: "However, we can exclude a major contribution of pollen-derived INP, because exposure to 90 °C deactivated all INP-8 in nine of the 10 samples treated that way and 92% of INP-8 in the remaining sample (26.3.-02.4.2014). Still, a few heat resistant INP-8 might have escaped observation at concentrations below the limit of detection in our approach (0.08 m-3). If so, they would on average not have constituted more than 7% of all INP-8 found prior treatment." (page 6, lines 26-29).*

```
4) It is hard to understand why the authors consider that "INP-8 may
consist largely of fungal spores during the warm part of the year
(Page 1, Line 15)" and "bacterial contributions may be more important
than fungal sources during the colder part of the year (Page 7, Line
25, 26)". I assume that this is only based on the results that "from
mid-May to mid-September, INP-8 correlated positively with the fungal
markers arabitol and mannitol (Page 1, Line 15; Page 6, Lines 26-
27)". This result may suggest that fungal spores are a potential
important source of INPs from mid-May to mid-September, but I think
that it is still impossible to rule out the possibility that the
contribution of bacteria and soil organic particles were also
significant. Did you try any other approaches to support the authors'
hypothesis? For example, did you evaluate the relationship of INP-8
with other markers (e.g., bacteria, soil organic particles) from mid-
May to mid-September?
```

Your interpretation of our views in the initial manuscript are correct. Reflecting on your comment and reading new papers we agree that the evidence we have at hand to support that view is not as strong as we initially thought it was. We do not have data for other parameters. When searching for answers to your comment we came across an interesting paper showing how bacterial and fungal communities on decaying leaves evolve in a highly dynamical and interacting succession (Purahong et al., 2016), which makes a strong point for the INP-8 sources probably also being highly dynamic throughout the seasons.

*Accordingly, we added to the end of section 3.4 the two sentences: "Overall, the relative contribution of $INP_{-8}$ from any type of microorganism might have changed by the end of September as a result of leaves starting to be shed by deciduous trees. Decaying leaves provide the substrate for a highly dynamical succession of interacting fungal and bacterial populations (Purahong et al., 2016)." (page 8, lines 6-8).*
*We also changed the statement cited in your comment, from "$INP_{-8}$ may consist largely of fungal spores during the warm part of the year (Page 1, Line 15)" to "some fraction of $INP_{-8}$ during that period may consist of fungal spores" (page 1, line 16); and changed "bacterial contributions may be more important than fungal sources during the colder part of the year" to "it might be that the fungal composition had changed or that bacteria had become more important sources of $INP_{-8}$." (page 8, line 3).*

```
5) It is a little hard to understand that "from mid-May to mid-
September, INP-8 correlated positively with the fungal markers
arabitol and mannitol (Page 1, Line 15)" only based on Figure 5. For
example, could you show the results (additional figures like Figure 4
or tables) comparing INP-8 with arabitol and mannitol measured in
different seasons (e.g., the period of mid-May to mid-September vs.
other periods; spring vs. summer vs. autumn vs. winter)?
```

*We have added the requested Figure (Figure 6).*

```
Technical corrections:
6) Page 2, Line 5: [f]ollow-up => follow-up?
```

*Changed as suggested.*

```
7) Please explain the definition of INP-8 more clearly. Is it ice
nucleating particles "active at -8 °C or warmer (Page 1, Line 25)" or
"active at -8 °C"?
```

Done.

*The sentence now reads: "ice nucleating particles active at -8 °C or at warmer temperatures (from here on collectively denominated as $INP_{-8}$)" (page 1, lines 26-27).*

```
8) It is a little difficult to see the data on precipitation and/or
snow depth in Figures 2 and 5. I would like to suggest that the
values would increase from bottom to top if there are no special
reasons.
```

We have tried the suggested change and found the Figure looks too crowded to properly capture its essence: the dynamics of INP mirror those of rainfall. By "mirroring" the precipitation and snow depth values on the horizontal axis (i.e. increasing from top to bottom) we find this feature is easier to capture.

*We added to the legend of Figure 2 the brief explanation "inverse scale to mirror precipitation values on the horizontal axis, disentangling an otherwise crowded display of the data" (page 13, lines 8-9).*

```
9) What is the value of 0.47 in Figure 4 (r or r2)? In addition, I
would like to suggest that the authors would indicate the equation of
the regression line, since they noted that "in PM2.5 we found
consistently about half as many INP-8 as in PM10 (Page 1, Line 14)".
```

The value 0.47 denotes the slope of a regression line fitted to the data (i.e. there were on average 0.47 times as many $INP_{-8}$ in $PM_{2.5}$ as there were in $PM_{10}$). The regression has an r value of 0.90.

*Changed Figure and its legend accordingly. (page 14).*

```
10) Is Figure 3 FLEXPART output? If so, please describe it in Section
3.2 and/or the caption of Figure 3. In addition, I would like to
suggest including the explanation of why the unit of potential
emission sensitivity is seconds (Page 4, Lines 3-4).
```

Yes, Figure 3 is FLEXPART output. The Figure legend is now more informative.

*Changed Figure legend to: "Source receptor sensitivity (SRS) fields for situations with > 4 $INP_{-8}$ $m^{-3}$ (left) and when $INP_{-8}$ were < 4 $m^{-3}$ (right) as derived from FLEXPART. The SRS unit is seconds, which would result in a mass concentration ($kg\ m^{-3}$) at the receptor when multiplied with an emission flux ($kg\ m^{-3}\ s^{-1}$) into the model grid cells. Since emission fluxes are not known for $INP_{-8}$, SRS values can be considered as a measure of relative impact that INP emissions from a particular area would have had on INP concentrations at Birkenes. The potential influence was strongest from areas shown in red colour and weakest from those in white and purple colours." (page 14, lines 3-8).*

*References*
*Bauer, H., Claeys, M., Vermeylen, R., Schueller, E., Weinke, G., Berger, A., and Puxbaum, H.: Arabitol and mannitol as tracers for the quantification of airborne fungal spores, Atmos. Environ., 42, 588–593, doi:10.1016/j.atmosenv.2007.10.013, 2008.*
*Burshtein, N., Lang-Yona, N, and Rudich, Y., Ergosterol, arabitol and mannitol as tracers for biogenic aerosols in the eastern Mediterranean, Atmos. Chem. Phys., 11, 829-839, doi:10.5194/acp-11-829-2011, 2011.*
*Purahong, W., Wubet, T., Lentendu, G., Schloter, M., Pecyna, M. J., Kapturska, D., Hofrichter, M., Krüger, D., and Buscot, F.: Life in leaf litter: novel insights into community dynamics of bacteria and fungi during litter decomposition, Mol. Ecol., 25, 4059-4074, 2016.*
*Weete, J. D., Abril, M., Blackwell, M.: Phylogenetic distribution of fungal sterols. PLoS ONE, 5, e10899, doi:10.1371/journal.pone.0010899, 2010.*
*Zhu, C., Kawamura, K., Fukuda, Y., Mochida, M., and Iwamoto, Y.: Fungal spores overwhelm biogenic organic aerosols in a midlatitudinal forest, Atmos. Chem. Phys., 16, 7497-7506, doi:10.5194/acp-16-7497-2016, 2016.*

MS No.: acp-2017-285

Response to JA Huffman (Referee)
* * *
To facilitate reading we use different fonts for `(1)comments from Referee,` (2) our response, *(3) the changes we made to the manuscript.*
* * *
`Conen et al. submitted a manuscript for review titled "Rainfall drives atmospheric ice nucleating particles in the maritime climate of Southern Norway." The manuscript compares 15 months of measurements of ice nucleating particles (at -8C), two molecular tracers (arabitol and mannitol), and rainfall data to present observations about INP behavior. The authors suggest that INP were likely to have local sources and are linked to rainfall, because of the evidence that INP concentrations correlated with rain. Further, they state that correlations with molecular tracers suggest INP "may consist largely of fungal spores." The manuscript presents interesting environmental data linking warm temperature INP with rainfall and two commonly used molecular tracers. The region sampled is also not over-represented in literature and so provides some atmospheric perspective on this region, including possible parallels with other similar regions of the world, as mentioned. In general, I support the publication of this manuscript, but there is some work that I suggest be done before it is accepted. The analysis and evidence that the observed INP are fungal in nature are both relatively thin and should be improved. I list some suggestions for specific additions below, including some possibilities for added discussion and some suggestions to add to quantitative evidence. These statements are meant to suggest possible areas of improvement, but are not necessarily meant to be comprehensive.`

We thank Alex Huffman for having taken the time to read our manuscript. His views on our work and his suggestions to improve it are much appreciated. Thank you also for drawing our attention to additional papers in this field.

The claim about the importance of fungal spores seems not enough supported by data. Also Referee #1 has made this point. We have reflected on this issue, read more papers and came to a more differentiated conclusion.

`General comments: Abstract: "INP(-8) correlated positive with the fungal markers arabitol and mannitol, suggesting that INP may consist largely of fungal spores." I think the confidence implied by this conclusions is somewhat over-stated. The evidence shown suggests to me that the INP have a source that is at least correlated with arabitol and mannitol, but this does not necessarily mean that the INP are spores themselves. The observations could also be explained if INP and fungal spores are co-emitted by a similar process or are somehow physically related to one another. A lot of evidence has suggested many fungal spores are not IN-active (e.g. Iannone et al., 2011) while other (i.e. rust spores from the cited Morris et al. paper) are IN-active at high temperatures. There is enough complexity in this conversation, that I think some discussion of these differences should be mentioned and the overall confidence in the implications that fungal spores are the source of INP should be scaled down a bit.`

We agree that the issue is more complex than initially portrayed in the manuscript.

*Accordingly, we added to the end of the 2nd paragraph of section 3.4: "However, spores found in the atmosphere are not only produced by rusts. For example, Cladosporium species contribute a large proportion of airborne spores (Maninnen et al., 2014) and their onset of freezing is well below -25 °C (Iannone et al., 2011). At the same time, some fungi release INP-8 from their mycelium in form of macromolecules (Fröhlich-Nowoisky et al., 2015), which are unlikely to contain storage products such as arabitol or mannitol." (page 7, lines 29-31).*

*The statement in the abstract is scaled down to "From mid-May to mid- September, INP-8 correlated positively with the fungal spore markers arabitol and mannitol, suggesting that some fraction of INP-8 during that period may consist of fungal spores." (page 1, lines 15-16). Further, a similar statement in the 1st paragraph of section 3.4 now reads: "These correlations (Fig. 6) support the above-mentioned notion (section 3.3) that FBAP and INP-8 may to some extent be of fungal origin, at least during the warmer part of the year." (page 7, lines 13-14).*

```
The discussion about molecular tracers should also be extended
somewhat. For example, arabitol and mannitol are commonly used as
tracers for fungal spores, but not without complications. One
important thing to mention here is that these specific tracers are
typically used as tracers for wet-discharge spores, but only poorly
relate to other types of spores (i.e. dry discharge spores like
Cladosporium that can be ubiquitous and a large fraction of spore
mass). How might this understanding impact the conclusions that are
being drawn here? I am aware that the general knowledge linking these
tracers with ice activity is low, and so it is unreasonable to
require any kind of a quantitative link between known ice fungal ice
nucleators and the amount of arabitol or mannitol they release, but I
suggest at least mentioning some of the uncertainties that come along
with the analysis and assumptions as presented.
```

Right, arabitol and mannitol are typical tracers for wet discharge spores (e.g. Zhu et al., 2016) and are not always well related to the primary fungal membrane sterol marker ergosterol (Burshtein et al., 2011). Another uncertainty are the dynamics in microbial succession which occur, for example, when leaves are shed and decay in autumn (Purahong et al., 2016). These issues call for an adaptation of the conclusions.

*We added to the beginning of section 3.4: "Arabitol and mannitol serve as carbohydrate stores in fungal spores. Their ambient air concentration has been found to correlate well with number concentrations of airborne fungal spores (Bauer et al., 2008), but not necessarily with ergosterol (Burshtein et al., 2011), a dominant sterol in most fungi (Weete et al., 2010). It seems that arabitol and mannitol are specifically associated with spores released under moist conditions, as occur in forests during nighttime (Zhu et al., 2016)." (page 7, lines 6-9).*
*To the end of the section we added: "Overall, the relative contribution of INP-8 from any type of microorganism might have changed by the end of September as a result of leaves starting to be shed by deciduous trees. Decaying leaves provide the substrate for a highly dynamical succession of interacting fungal and bacterial populations (Purahong et al., 2016)." (page 8, lines 6-9).*
*We changed the second paragraph of the Conclusions. It now reads: "The assumption of relevant fungal sources is supported during the warmer part of the year by some similarities in the temporal pattern of INP-8 and the fungal spore markers arabitol and mannitol. However, major shifts in microbial community composition occur when leaves are shed in autumn and start to feed a highly dynamical succession of interacting fungal*

*and bacterial populations. These dynamics likely also affect the strength and composition of the various sources of INP$_{-8}$." (page 8, lines 18-22).*

In general, I think Section 3.2 and Figure 3 need more detailed
discussion and explanation to help a reader not experienced with this
type of analysis. Can you explain what the z-scale implies from this
figure and how it relates to the brief observations you make? It's
hard to know how much to make of the summary observations reported.
How much of this is a function of different averaging times that may
lead to random differences? If this is an important piece of
evidence, is there some statistical treatment that can be applied
here? Flipping back and forth between the comments and the figure I
can follow the reasoning of the trends mentioned, but it is hard to
know whether the "striking" comment (P5 L14) is stronger than I would
have stated — at least having briefly looked at the differences. If
the authors are confident of the strong difference, I suggest
improving the evidence for that distinction. In contrast to this,
however, the last sentence in Section 3.2 essentially says that the
authors think the effects are local, which implies to me that Figure
3 should provide evidence *against* long-distance sources, right?
This goes back to how Figure 3 should be interpreted as striking
differences between high and low INP concentration.

We summarised the source receptor sensitivity (SRS) fields (also called "footprints") for higher (>4 m$^{-3}$) and lower (<4 m$^{-3}$) concentrations of INP-8 in order to see whether the higher INP concentrations coincided with "footprints" extending to a region that did not have an influence when concentrations were low (or vice versa). The "footprints" shown in Figure 3 are cumulative, i.e. the sum of all "footprints" for weeks with either high or low concentrations of INP. The analysis here is qualitative in the sense that we look for differences in general patterns. There were no additional areas associated with high INP concentrations which speaks against long distance transport, at least long distance transport from a certain area. The greatest difference is a lack of influence from the north-eastern quadrant when concentrations are low. We had initially looked at "footprints" averaged over one week, where we could also see the lack of influence from the north-eastern quadrant for weeks with INP <4 m$^{-3}$, but no general pattern in the other quadrants. Summarising the pattern of each concentration category provides for a more direct visual access to this information.

*We tried to make the interpretation more accessible by expanding the 2$^{nd}$ sentence in section 3.2: "Higher concentrations of INP$_{-8}$ were not associated with source areas not seen when INP$_{-8}$ were < 4 m$^{-3}$. Hence, they were not transported to Birkenes from specific strong sources afar. The main difference when INP$_{-8}$ were > 4 m$^{-3}$ was a weaker influence from the northeast ..." (page 5, lines 24-26). This re-wording also replaced "strikingly less" with "weaker".*
*Further, the z-scale is now explained in the Figure legend: "Figure 3: Source receptor sensitivity (SRS) fields for situations with > 4 INP$_{-8}$ m$^{-3}$ (left) and when INP$_{-8}$ were < 4 m$^{-3}$ (right) as derived from FLEXPART. The SRS unit is seconds, which would result in a mass concentration (kg m$^{-3}$) at the receptor when multiplied with an emission flux (kg m$^{-3}$ s$^{-1}$) into the model grid cells. Since emission fluxes are not known for INP$_{-8}$, SRS values can be considered as a measure of relative impact that INP emissions from a particular area would have had on INP concentrations at Birkenes. The potential influence was strongest from areas shown in red colour and weakest from those in white and purple colours." (page 14, lines 3-8).*

What happens if you do correlations of the traces in Figures 2 and 5?
A lot of the observations come down to qualitative comparisons of

these traces, but it is hard to know what this means quantitatively.
I think this is one obvious area that could easily improve the
manuscript. Without evidence beyond the visual trends presented, it
is hard to know how much to make of the possible co-variance. As a
simple addition, I would also add the R2 value (or something similar)
to Figure 4.

*Figure 4 has been adapted as suggested (page 14) and we also added a Figure showing INP$_{-8}$ plotted against the fungal tracers, with different symbols for the warmer and the colder parts of the year (Figure 6, page 15).*

Looking at Figure 5, it seems that there is a one-point lag in INP
behind arabitol and mannitol during approximately October and
November. Do you think this is real? If so, what might be causing
this? Or is it just a figure illusion and statistical aberration.

The ups and downs in the concentration of both tracers run in parallel throughout October and November. The impression of a time lag might be caused by the concentration of arabitol having dropped to a very small value (2.6 ng m$^{-3}$) on 15. November, while that of mannitol had only dropped to an intermediate value (13.0 ng m$^{-3}$). The following week they both drop again, although mannitol by a much larger fraction. After that they change almost synchronously.

P7 L10: The statement here is that "Since INP were no longer directly
related to fungal spore markers during this period, it might be that
bacteria contributed more to the total number . . .". Another
possibility is that the type of spores being released are of a
different variety and are just less efficient at producing IN-active.

We agree that there are alternative explanations for this observation.

*The sentence now reads: "Since the time course of INP$_{-8}$ was no longer directly related to that of fungal spore markers during this period, it might be that the fungal composition had changed or that bacteria had become more important sources of INP$_{-8}$." (page 8, lines 2-3).*

Bigg, Soubeyrand, and Morris recently published a paper reporting
long-term statistical correlations between ice nuclei and rainfall in
Australia (Bigg et al., 2015). I think reference to this work would
be appropriate here, probably in the conclusions.

*We have added the reference in the Introduction (page 2, line 4).*

P6 L1: The authors cite Schumacher et al. as having observed a 2.5 —
3.0 um mode of fluorescent particle during 18-months of study in
Finland. That paper also mentions a prominent decrease in fluorescent
particles during snow-covered periods, which qualitatively matches
some of the observations shown here.

*Made reference to this observation on page 6, lines 9-10.*

Is snowfall poor at launching INP because of snow-covered vegetation
and soil or also because the kinetic velocity at which the drops fall
does not kick up material? Some recent papers on rainfall velocity
and particle ejection could be cited and discussed here (e.g. P7
L18).

We think that most of the difference comes down to the lower deposition velocity, hence lower kinetic energy, of snowflakes.

*Added to the end of the 1st paragraph of section 3.1: "Deposition velocity of snow is in the order of 1 m s$^{-1}$ (Garrett et al., 2012), that of even a small raindrop (1 mm diameter) is already four times as large (Gun and Kinzer, 1949). Hence, for the same mass, the kinetic energy of rain (proportional to the velocity squared) is at least an order of magnitude larger compared to precipitation in form of snow, and so the energy is available for dispersion and aerosolisation of particles." (page 4, lines 24-27).*

```
P7 L23: Are the heat treatment properties of fungal proteins the same
as bacterial proteins? I think of spores as relatively robust, and so
I wonder if it is possible for some fraction of spore material to
remain active, whereas the fraction for bacteria goes to zero? In any
case, I think the evidence for these arguments should be stronger.
```

They are not exactly the same. Evidence for the heat sensitivity of bacterial and fungal INP summarised by Pummer et al. (2015) shows that bacterial INP are deactivated already by moderate heat (40 °C), whereas some fungal INP tolerate temperatures close to 60 °C and few close to 100 °C. All we can say from our observations is that most of the INP seen prior treatment were either of bacterial or fungal origin because >93% of the INP in our samples were deactivated at 90 °C.

*Added the reference to Pummer et al. (2015) at the corresponding sentence (page 6, lines 29-30): "The INP of bacteria and most fungi are proteins and denatured at this temperature (Pummer et al., 2015)."*

```
P 6 L12: Another paper by Maninnen et al. (2014) shows seasonal
trends in pollen and fungal spores at the boreal Hyttiala site in
Finland and they also break the analysis down into PM mass <2.5 um
and >10 um. While not at the same land-use type, these measurements
may (or may not) be useful for broad comparison here.
```

Thank you for the reference.

*In the revised manuscript it lends support to two statements, one about pollen as an important source of FBAP (page 6, line 22), the other about the large proportion of Cladosporium in airborne spores (page 7, line 29).*

Minor technical comments: P1 L10: Move placement of "probably" to "INP were probably aerosolized . . ."

*Done.*

P1 L12-14: I though this sentence was confusing and could use a revision to make the point clearer.

*Changed sentence to: "Further, transport model calculations for large (> 4 m-3) and small (< 4 m-3) numbers of INP-8 revealed greater differences in the likelihood of the potential source regions to provide precipitation to Southern Norway, than in the proportion of land cover or land use type." (page 1, lines 12-14).*

P1 L22: snowflake is one word P4 L7 , L8, L10: "Landuse" should be two words P5 L19: Specifically mention that Tenerife is off the W coast of northern Africa.

*Done.*

*Correspondence to*: Franz Conen (franz.conen@unibas.ch)

**Abstract.** Ice nucleating particles active at modest supercooling (e.g. -8 °C; $INP_{-8}$) can transform clouds from liquid to mixed-phase, even at very small number concentrations (<10 $m^{-3}$). Over the course of 15 months, we found very similar patterns in weekly concentrations of $INP_{-8}$ in $PM_{10}$ (median = 1.7 $m^{-3}$, maximum = 10.1 $m^{-3}$) and weekly amounts of rainfall (median = 28 mm, maximum = 153 mm) at Birkenes, Southern Norway.  Most $INP_{-8}$ were probably aerosolised locally by the impact of raindrops on plant, litter and soil surfaces. Major snowfall  and heavy rain onto snow-covered ground were not mirrored by enhanced numbers of $INP_{-8}$. Further,  transport model calculations for large (> 4 $m^{-3}$) and small (< 4 $m^{-3}$) numbers of $INP_{-8}$ revealed greater differences in the likelihood of the potential source regions 
[revised manuscript text omitted]

Arabitol and mannitol have previously been identified as amenable tracers of fungal spores (Bauer et al., 2008). Concentrations of arabitol and mannitol in $PM_{10}$ filter samples were determined using Waters Acquity ultra-performance liquid chromatography (UPLC) in combination with Waters Premier XE high-resolution time-of-flight mass spectrometry (HR-TOFMS) operated in the negative electrospray ionization (ESI-) mode: resolution > 10000 FWHM (Full width half maximum). The analytical methodology is based on that described by Dye and Yttri (2005) for monosaccharide anhydrides, deviating from the original one only by choice of the column (2.1 x 150 mm HSS T3, 1.8 µm, Waters Inc.). Arabitol and mannitol were identified on the basis of retention time and mass spectra of authentic standards (ICN Biomedicals). Response factors for arabitol and mannitol were calculated from external standards. Isotope-labelled standards of levoglucosan ($^{13}$C-levoglucosan, 98%, Cambridge Isotopic Laboratories) were used as internal recovery standard.

**2.5 FLEXPART**

[revised manuscript text omitted]

Another important source of FBAP in PM$_{10}$ are pollen (Manninen et al., 2014). Pollen from birch, the most abundant deciduous tree around the Birkenes Observatory, has an aerodynamic diameter of 20 μm (Efstathiou et al., 2011). Smaller fragments of pollen are generated by osmotic rupture when pollen grains get wet. These fragments are as IN-active as intact grains (Pummer et al., 2012). On rainy days their abundance increases in the fine fraction (Rathnayake et al., 2017). However, we can exclude a major contribution of pollen-derived INP, because exposure to 90 °C deactivated all INP$_{-8}$ in nine of the 10ten samples treated that way and 92% of INP$_{-8}$ in the remaining sample (26.3.-02.4.2014). Still, a few heat resistant INP$_{-8}$ might have escaped observation at concentrations below the limit of detection in our approach (0.08 m$^{-3}$). If so, they would on average not have constituted more than 7on average > 93% of all INP$_{-8}$ found prior to treatment in our assays. The INP of bacteria and most fungi are proteins and denatured at this temperature (Pummer et al., 2015). INP from

**Comment [FC1]:** (average of lower limit of detection (0.08 m3) / number of INP observed prior heat treatment)

pollen or fractured pollen are most likely polysaccharides and would have remained active after heating to 90 °C (Pummer et al., 2012, 2015).

Recently, Wang et al. (2016) described a mechanism that generates airborne soil organic particles (ASOP) of sub-micron size by air bubbles bursting at the water-air interface of impacted raindrops. Soil organic matter can harbour large numbers of INP$_{-8}$ (Schnell and Vali, 1972; Conen et al., 2011; O'Sullivan et al., 2015; Hill et al., 2016). Therefore, some of the INP$_{-8}$ in the fine fraction at Birkenes might be of that kind.

**3.4 Time series of arabitol and mannitol**

Arabitol and mannitol serve as carbohydrate stores in fungal spores. Their ambient air concentration has been found to correlate well with number concentrations of airborne fungal spores (Bauer et al., 2008), but not necessarily with ergosterol (Burshtein et al., 2011), a dominant sterol in most fungi (Weete et al., 2010). It seems that arabitol and mannitol are specifically associated with spores released under moist conditions, as occur in forests during nighttime (Zhu et al., 2016). serve for carbohydrateambient air At Birkenes, aArabitol and mannitol had very similar temporal pattern throughout the year (Fig. 5). Their concentrations were low from January to mid-April, then increased, and remained enhanced throughout summer. During the warmer part of the year, from mid-May to mid-September, they correlated significantly with INP$_{-8}$ (mannitol: r = 0.72, p < 0.01; arabitol: r = 0.48, p < 0.05). These correlations (Fig.ure 6) support the above-mentioned notion (section 3.3) that FBAP and INP$_{-8}$ may be mainlyto some extent be of fungal origin, at least during the warmer part of the year. Another hint in this direction comes from measurements about 300 km north east of Birkenes. There, in summer, arabitol and mannitol had a unimodal size distribution peaking between 2 and 4 µm aerodynamic diameter (Yttri et al., 2007), which coincides with our interpretation of the 50-50% distribution of INP$_{-8}$ amongst PM$_{2.5}$ and PM$_{10-2.5}$. Concentrations of the fungal spore markers increased towards the end of three rather dry weeks in September and reached their annual maxima with intensive rainfall at the beginning of October, followed one week later by a sevenfold increase in INP$_{-8}$. Changes in INP$_{-8}$ continued to follow those of the fungal fungal spore markers with a delay of one week until the beginning of December. During In December, INP$_{-8}$ remained elevated while whereas concentrations of arabitol and mannitol decreased markedly. The fungal spore markers and INP$_{-8All}$ reached their minimum with snowfall in the last week of the year.

Assuming 1.2 pg arabitol and 1.7 pg mannitol per fungal spore (Bauer et al., 2008), we estimate for the period from mid-May to mid-September average spore concentrations of 4.6 and 5.8 x 10$^3$ m$^{-3}$. The average concentration of INP$_{-8}$ during the same period was 1.6 m$^{-3}$. If all INP$_{-8}$ were spores, there would have been 2.8 or 3.5 x 10$^{-4}$ INP$_{-8}$ per spore, which is in the upper range of values reported by Morris et al. (2013, Fig. 1) for urediospores of rusts. However, spores found in the atmosphere are not only produced by rusts. For example, *Cladosporium* species contribute a large proportion of airborne spores (Maninnen et al., 2014). T and their onset of freezing is well below -25 °C (Iannone et al., 2011). At the same time,

some fungi release INP$_{-8}$ from their mycelium in form of macromolecules (Fröhlich-Nowoisky et al., 2015), which are unlikely to contain storage products, such as arabitol or mannitol.

For the last three months of the year, the average INP$_{-8}$ to spore ratio was about twice as large as compared to the preceding four months. Since the time course of INP$_{-8}$  was no longer directly related to that of fungal spore markers during this period, it might be that the fungal composition had changed, or that bacteria had became more important sources of INP$_{-8}$ . The expression of ice nucleation activity in bacteria is favoured by cold temperatures and nutrient limitation (Nemecek-Marshall et al., 1993). Hence, when similar numbers of bacteria are aerosolised by the same amount of rainfall, they likely contribute larger numbers of INP$_{-8}$ during the cold season, than during the warm months. Overall, the relative contribution of INP$_{-8}$ from any type of microorganism might have changed by the end of September, as a result of leaves starting to be shed by deciduous trees. Decaying leaves provide the substrate for a highly dynamical succession of interacting fungal and bacterial populations (Purahong et al., 2016).

**4 Conclusion**

Abundant rainfall in the  coastal climate of Southern Norway drives the near surface concentration of INP$_{-8}$ across all seasons. Concentrations of INP$_{-8}$ increase with amounts of rain. Most airborne INP$_{-8}$ are probably aerosolised locally through the impact of raindrops onto surfaces hosting microorganisms that synthesise INP. Snowfall has no such effect. When trees are defoliated between October and April, decaying leaf litter on the ground constitutes a likely INP source. During this time, snow cover on the ground strongly reduces such INP aerosolisation by rainfall.

Rain-released INP$_{-8}$ are equally distributed amongst the fine (PM$_{2.5}$) and the coarse fraction (PM$_{10-2.5}$) of PM$_{10}$. Sensitivity to heat treatment (90 °C) suggests bacterial and fungal sources, (and) not pollen. The assumption of relevant fungal sources is  supported during the warmer part of the year by some similarities in the temporal pattern of INP$_{-8}$ and the fungal spore markers arabitol and mannitol. However, major shifts in microbial community composition occur when leaves are shed in autumn and start to feed a highly dynamical succession of interacting fungal and bacterial populations. These dynamics likely also affect the strength and composition of the various sources of INP$_{-8}$.

In general terms, we expect similar relations between rainfall and warm temperature INP in other coastal regions with a comparable climate and ecosystem, such as the Pacific coasts of Canada and Chile, Japan, and New Zealand. Global warming may lead to shorter periods of snow cover on the ground and a greater proportion of precipitation falling as rain

instead of snow. These changes would probably result in larger airborne concentrations of $INP_{-8}$ during the cold season. Whether they have an effect on cloud development in these regions remains an interesting question for further studies.

**5 Acknowledgements**

The $PM_{10}$ and $PM_{2.5}$ filter samples used for measurements of ice nucleating particles in the present study were analysed from filters obtained as part of the Norwegian national monitoring program (Aas et al., 2015). Precipitation and snow data were obtained from the Norwegian Meteorological Institute through eKlima. We thank Alex Huffman and a second, anonymous Reviewer for constructive comments leading to a more profound interpretation of our results. FLEXPART calculations were supported by Nordforsk, as part of the Nordic Centre of Excellence "eScience Tools for Investigating Climate Change at High Northern Latitudes" (eSTICC).

[Figure]

**Figure 1: Location of Birkenes Observatory  (left) and a view of the Observatory (right).**

[Figure]

**Figure 2: Time course of INP$_{-8}$ in the PM$_{10}$ (black) and PM$_{2.5}$ (green) particle size fractions. Peaks in INP$_{-8}$ largely mirror peaks in precipitation (blue; inverse scale to mirror precipitation values on the horizontal axis, disentangling an otherwise crowded display of the data). Snow (open circles) or small amounts of rain are accompanied by very small numbers of INP$_{-8}$ in the same week.**

10 **Significant amounts of snow on the ground (grey bars) seem to attenuate the increase in INP$_{-8}$ with intense rainfall. Total amount of precipitation in 2014 was 2077 mm (entire period shown: 2640 mm). Precipitation and snow cover data are the averages of three stations operated by the Norwegian Meteorological Institute in the municipality of Birkenes.**

[Figure]

**Figure 3: Source receptor sensitivity (SRS) fields for situations with > 4 INP$_{-8}$ m$^{-3}$ (left) and when INP$_{-8}$ were < 4 m$^{-3}$ (right) as derived from FLEXPART.** ~~Colours denote the relative impact that INP emissions from a particular area would have had on INP concentrations at Birkenes. The potential influence was strongest from areas in red and weakest from those in white and purple. Values are in an arbitrary unit of time, denoting the relative duration that a potential source at a specific point on the map would have had to influence concentrations observed at Birkenes.~~ The SRS unit is seconds, which would result in a mass concentration (kg m$^{-3}$) at the receptor when multiplied with an emission flux (kg m$^{-3}$ s$^{-1}$) into the model grid cells. Since emission fluxes are not known for INP$_{-8}$, SRS values can be considered as a measure of relative impact that INP emissions from a particular area would have had on INP concentrations at Birkenes. The potential influence was strongest from areas shown in red colour and weakest from those in white and purple colours.

[Figure]

**Figure 4: Ice nucleating particles active at -8 °C in PM$_{2.5}$, relative to their number in PM$_{10}$.**

[Figure]

**Figure 5: Weekly concentration of arabitol (red) and mannitol (magenta). The time courses of INP$_{-8}$ in PM$_{10}$ (black; x 5, to fit onto the same scale) and precipitation (blue) are copied from Fig. 1 to facilitate direct comparison.**

[Figure]

[Figure]

**Figure 6: Correlation between INP$_{-8}$ and the fungal markers mannitol (left) and arabitol (right) for the time period from 14. Mai to 24. September, 2014 (black circles, regression line and equation are fitted to these data). Concentrations of INP$_{-8}$ before and after this period (crosses) did not correlate with either fungal spore marker.**